# Treatment with Rasburicase in Hospitalized Patients with Cardiorenal Syndrome: Old Treatment, New Scenario

**DOI:** 10.3390/ijms25063329

**Published:** 2024-03-15

**Authors:** Rosa Melero, Beatriz Torroba-Sanz, Marian Goicoechea, Iago Sousa-Casasnovas, Jose María Barrio, Ana María García-Prieto, Patrocinio Rodriguez-Benitez, Xandra García-González, María Sanjurjo-Sáez

**Affiliations:** 1Nephrology Department, Hospital General Universitario Gregorio Marañón, Instituto de Investigación Sanitaria Gregorio Marañón (IiSGM), 28007 Madrid, Spain; mariarosa.melero@salud.madrid.org (R.M.); marian.goicoechea@salud.madrid.org (M.G.); agarciaprieto@salud.madrid.org (A.M.G.-P.); prodriguezb@salud.madrid.org (P.R.-B.); 2Pharmacy Department, Hospital General Universitario Gregorio Marañón, Instituto de Investigación Sanitaria Gregorio Marañón (IiSGM), 28007 Madrid, Spain; beatriz.torroba@salud.madrid.org (B.T.-S.); maria.sanjurjo@salud.madrid.org (M.S.-S.); 3Cardiology Department, Hospital General Universitario Gregorio Marañón, Instituto de Investigación Sanitaria Gregorio Marañón (IiSGM), 28007 Madrid, Spain; 4Anesthesia Department, Hospital General Universitario Gregorio Marañón, Instituto de Investigación Sanitaria Gregorio Marañón (IiSGM), 28007 Madrid, Spain; jbarriog@salud.madrid.org

**Keywords:** cardiorenal syndrome, hyperuricaemia, rasburicase, uricase, diuretic resistance, continuous renal replacement therapy, cardiac surgery, renal dysfunction, congestive nephropathy

## Abstract

Cardiorenal syndrome (CRS) involves joint dysfunction of the heart and kidney. Acute forms share biochemical alterations like hyperuricaemia (HU) with tumour lysis syndrome (TLS). The mainstay treatment of acute CRS with systemic overload is diuretics, but rasburicase is used in TLS to prevent and treat hyperuricaemia. An observational, retrospective study was performed to assess the effectiveness and safety of a single dose of rasburicase in hospitalized patients with cardiorenal syndrome, worsening renal function and uric acid levels above 9 mg/dL. Rasburicase improved diuresis and systemic congestion in the 35 patients included. A total of 86% of patients did not need to undergo RRT, and early withdrawal was possible in the remaining five. Creatinine (Cr) decreased after treatment with rasburicase from a peak of 3.6 ± 1.27 to 1.79 ± 0.83 mg/dL, and the estimated glomerular filtration rate (eGFR) improved from 17 ± 8 to 41 ± 20 mL/min/1.73 m^2^ (*p* = 0.0001). The levels of N-terminal type B Brain Natriuretic Peptide (Nt-ProBNP) and C-reactive protein (CRP) were also significantly reduced. No relevant adverse events were detected. Our results show that early treatment with a dose of rasburicase in patients with CRS and severe HU is effective to improve renal function and systemic congestion, avoiding the need for sustained extrarenal clearance, regardless of comorbidities and ventricular function.

## 1. Introduction

Cardiorenal syndrome (CRS) is defined as a disorder of the heart and kidneys whereby acute or chronic dysfunction in one organ can induce acute or chronic dysfunction in the other [1].

In humans, uric acid (UA) is the end product of the metabolism of purines, which are part of nucleotides. Hyperuricaemia (HU) is caused by inadequate renal excretion in about 90% of cases and is defined as an elevated serum uric acid level, usually greater than 6 mg/dL in women and 7 mg/dL in men.

There is increasing evidence that uric acid plays a role in the pathogenesis of CRS [2]. Elevated serum uric acid levels are a common finding in people with CRS with activation of the renin–angiotensin–aldosterone system (RAAS), especially if there is concomitant treatment with high-dose diuretics, calcineurin inhibitors, iodinated contrast agents or cardiac surgery [3,4,5,6,7]. HU can cause silent tissue damage and is considered an independent risk factor for the development and progression of kidney injury and cardiovascular disease [8,9].

It has been found that serum UA concentrations over 6.8 mg/dL bring on tissue precipitation by crystallization to monosodium urate [10]. Dehydration and acidic urinary pH also favour the precipitation of urate, obstructing the collecting ducts and increasing intrarenal pressure, with a reduction in blood flow and renal glomerular filtration rate (GFR) [11,12]. Tissues damaged by crystals act as a lure for the initiation of the inflammatory response and, ultimately, cell death [13,14,15]. In addition to obstruction and oxidative stress, UA causes endothelial dysfunction due to a reduction in nitric oxide, renal vasoconstriction, loss of autoregulation, activation of angiotensin II, reduction in glomerular filtration, hypertension, renal ischaemia and activation of the immune system [16,17,18,19,20,21]. The presence of HU in acute coronary syndrome, decompensated heart failure or cardiac surgery has also been associated with longer hospital stay, higher mortality from cardiovascular disease and all-cause mortality [22,23,24].

Acute forms of CRS share biochemical alterations of tumour lysis syndrome, such as elevated uric acid, phosphorus (P), potassium (K) and lactate dehydrogenase (LDH) [25]. In particular, we can find spontaneous or non-tumour lysis syndrome in decompensated cardiac patients with ventricular assist devices or in the postoperative period of cardiac surgery with haemolysis, rhabdomyolysis or hyperthermia, or after surgery with general anaesthesia [26].

In haematologic cancers, the treatment of TLS is based mainly on intensive volume expansion with 3–4 L/1.73 m^2^ of crystalloids, together with rasburicase (uricase) [26,27]. The aggressive fluid resuscitation maintains renal perfusion and glomerular filtration rate, increases urine flow rate, dilutes urine, and improves uric acid excretion. This prevents precipitation of UA due to intratubular accumulation and thus reduces the need for renal replacement therapy. Still, in those cases where there is concomitant renal failure or volume overload, the use of loop diuretics and renal replacement therapy (RRT) can be necessary.

In patients with CRS and systemic overload, intensive volume expansion is not feasible and, therefore, treatment is based on decongestion with high-dose diuretic therapy to achieve a diuresis target of about 3 L/day [28]. In many patients, this objective is difficult to achieve due to the presence of diuretic resistance and renal dysfunction, and in these cases, RRT is often necessary [29].

A small number of studies suggest that treating asymptomatic HU may have a positive impact not only on chronic kidney disease (CKD) progression, but also on cardiovascular and survival outcomes [8].

Rasburicase treatment is approved for the treatment and prophylaxis of acute hyperuricaemia in order to prevent acute renal failure in adults, children and adolescents (aged 0 to 17 years) with haematological malignancy with a high tumour burden and at risk of a rapid tumour lysis or shrinkage at initiation of chemotherapy. Rasburicase has a rapid effect in 4 h, transforming uric acid into allantoin, a metabolite easily eliminated through urine, and resulting in an increase in urinary flow [30,31,32,33].

Our hypothesis was that in patients with cardiorenal syndrome and severe hyperuricaemia, treatment with rasburicase may improve renal function, avoid acute tubular necrosis and avoid the need for renal replacement therapy (RRT). Therefore, the aim of this study was to assess the effectiveness and safety of a single dose of rasburicase in hospitalized patients with cardiorenal syndrome and uric acid levels above 9 mg/dL.

## 2. Results

Thirty-five patients with a mean age of 67 ± 15 years (23 males and 12 females) admitted to different cardiology units and treated with rasburicase were included. Eighteen patients were hospitalized in clinical cardiology and seventeen were hospitalized in a cardiac critical care unit (the coronary care unit (CCU) or the post-cardiac surgery intensive care unit (CICU)).

Baseline and analytical characteristics of the 35 patients are summarized in Table 1.

All patients were receiving loop diuretics, orally (14.2%) or intravenously (85.7%). A total of 48.6% of the patients were receiving furosemide in continuous perfusion at a concentration of 2.5 mg/mL. Among the thirty-five patients, sixteen (45.7%) were receiving a combination of two or more diuretics.

Over half of the patients had received a potentially nephrotoxic drug during admission, and around 40% of the patients were receiving inotropic and/or vasopressor drugs.

A total of 37.14% of patients were receiving xanthine oxidase inhibitors as part of their usual treatment.

The patients received a single dose of intravenous rasburicase according to the active hospital protocol at the time of administration. Accordingly, the first nine patients received a dose of 0.20 mg/kg, and the following patients received a total dose of 6 mg.

At the time of rasburicase administration, median UA levels were 12.9 ± 2.5 mg/dL. Except in one case, UA was reduced to <0.1 mg/dL in less than 24 h. In this patient, UA was reduced from 16.1 mg/dL to 5.2 mg/dL, but a diuretic response of 4490 cc was achieved.

No clinically relevant side effects were reported in relation to the rasburicase treatment. In one patient, Coombs-positive haemolytic anaemia of unknown origin was detected prior to administration of rasburicase. It was not considered clinically relevant; it did not worsen after rasburicase administration, and no transfusions of blood products were needed.

### 2.1. Renal Fuction

The mean baseline Cr and eGFR were 1.58 ± 0.68 mg/dL and 43.41 ± 19 mL/min/1.73 m^2^, respectively. During the hospitalization episode, the mean values for the peak Cr and lowest eGFR were 3.6 ± 1.27 mg/dL and 17.5 ± 8.7 mL/min/1.73 m^2^, and 48–72 h after rasburicase administration, the mean Cr decreased to 2.5 ± 1.0 mg/dL. At discharge, the mean Cr improved to 1.8 ± 0.83 mg/dL, with an eGFR of 42 ± 21.7 mL/min/1.73 m^2^ (*p* < 0.0001) (Figure 1).

All patients (100%) achieved diuretic response between 12 and 98 h after rasburicase administration, with a mean diuresis of 2800 ± 1200 cc/day compared to a mean diuresis of 1540 ± 620 cc/day (*p* < 0.0001) before rasburicase administration. In those patients receiving IV furosemide combined with other diuretics (34.3%), the mean diuretic response after rasburicase administration was 4.200 ± 1051 cc/day.

Five patients (14.2%) required RRT by continuous venovenous haemofiltration (CVVH) for a maximum of 98 h. All patients recovered renal function.

The patients requiring RRT were younger (54 ± 17 vs. 70 ± 14 years; *p* = 0.03) and had higher CRP levels (80.5 (1.6–186) vs. 10.9 (3.5–29.4) mg/L), although this difference was not statistically significant. There were no significant differences in basal renal function, uric acid, phosphorus, potassium or sodium bicarbonate levels between both groups of patients. We also found no statistically significant association between the need for RRT and comorbidities such as DM, liver disease, solid organ transplantation or the need for a VAD.

Renal biopsies were performed in two patients for clinical reasons not related to CRS (Figure 2). No intratubular or interstitial uric acid deposits or inflammatory infiltrates were detected. The first biopsy showed thickening of small-calibre arteries at the expense of concentric myointimal hyperplasia and subintimal fibrosis, with occlusion of the arteriolar lumen of up to 60–70%, causing ischaemic changes in the glomerular tuft. Moderate arteriolar hyalinosis was also found. Some glomeruli showed marked hyperplasia of the juxtaglomerular apparatus, consistent with the clinical diagnosis of renovascular hypertension. The second biopsy showed small-calibre vessels with intimal fibroelastosis with occlusion of 50% of the vascular lumen, in addition to multifocal arteriolar hyalinosis. Regenerative changes and tubular dilation with focal loss of the brush border in its epithelium can be seen to fit a diagnosis of benign nephroangiosclerosis with signs of acute tubular necrosis.

### 2.2. Cardiac Function

Nt-ProBNP was reduced after rasburicase administration from a median of 24,293 to 6034 pg/mL *p* < 0.0001 at hospital discharge (Table 2).

In patients who required RRT compared to those who did not, we found no differences in levels of Nt-proBNP or lactic acid, nor in markers of left and right ventricular dysfunction. We also found no significant differences between patients who required controlled ventilatory support (CVS) and RRT (see Table 3).

All patients requiring RRT were treated with inotropic drugs. Among those patients not requiring RRT, the percentage of patients receiving inotropics was much lower (33%), and this difference was statistically significant (*p* = 0.009).

### 2.3. Inflammation and Mortality

The median baseline CRP was 10 (3.4–39.2) mg/L, with a significant reduction at discharge to 4.3 (1.1–10.2) mg/L (*p* < 0.0001).

Four patients (11.3%) died during admission: one patient died due to COVID-19 infection, another due to septic shock, and two of terminal chronic heart failure. Two of these patients had required RRT for more than 48 h, but both had recovered diuretic rhythm with improvement of renal function prior to exitus.

## 3. Discussion

This is the first study to evaluate the use of rasburicase in CRS and severe HU. Our results indicate that early treatment with a single dose of rasburicase improves renal function in the short term, independently of comorbidities, drugs received and ventricular function, improving systemic congestion and avoiding the need for sustained extra-renal clearance, with no clinically relevant adverse events.

Renal disease is present in up to 91% of patients admitted for acute heart failure, and the presence of HU is considered an independent risk factor for renal dysfunction [34,35]. Systemic congestion is one of the main causes of CRS, with congestive nephropathy (CN) being the term that denotes renal involvement in congestion [36]. There is some similarity between CN in CRS and the uric acid nephropathy of TLS. In CN, increased central venous pressure is transmitted to the renal vein, raising the interstitial hydrostatic pressure, whereas in uric acid nephropathy, it is the obstruction that increases intratubular and peritubular capillary pressure [37,38,39]. Elevation of intrarenal pressure reduces blood flow and GFR, activates the renin–angiotensin–aldosterone system (RAAS), causes vasoconstriction of the afferent arteriole, reduces nitric oxide (NO), and activates the inflammatory response [12,40]. In CRS, other clinical conditions may favour the development of hyperuricaemia and may be considered as alarm signs such that a determination of uric acid should be encouraged (Table 4).

In our cohort, 37.4% of the patients were receiving xanthine oxidase inhibitors (XOis) as part of their chronic treatment. Although they could potentially help prevent severe hyperuricaemia, XOis exert their action by inhibiting the de novo formation of uric acid, thus maintaining the renal toxicity of the xanthine crystals that are formed by inhibiting xanthine oxidase, whereas rasburicase is capable of eliminating the preformed uric acid, avoiding its deleterious effects and preventing the formation of xanthine crystals. On the other hand, uricosuric drugs (e.g., lesinurad, benzbromanone and probenecid) are contraindicated in patients with severe renal failure. For this reason, we decided to treat our patients with rasburicase and not use xanthine oxidase inhibitors or uricosurics in the acute phase of CRS.

In regard to hyperuricaemia, SGLT2 inhibitors can also play a relevant role. In patients with type 1 diabetes, Lytvyn et al. found that by increasing glucosuria, SGLT2i can provide a 10 to 15% reduction in plasma urate because of uric acid secretion in exchange for glucose reabsorption via the GLUT9 transporter [41]. This possible relationship between plasma uric acid levels and SLGT2-I-associated uricosuria may be beneficial to prevent CRS, although clinical relevance is not yet established. However, the initiation of SGLT2i is not currently recommended in situations of acute renal dysfunction. Manufacturers advise that dapagliflozin treatment should not be initiated in patients with GFR < 25 mL/min and that empagliflozin should not be initiated in patients with GFR < 20 mL/min, due to limited experience. Their relevance to CRS should be addressed in future research. On the other hand, rasburicase can be used safely in acute renal failure to prevent further deterioration.

Rasburicase is used for the treatment and prophylaxis of acute hyperuricaemia associated with TLS. The conversion of uric acid to allantoin increases the diuresis rate and improves renal function, decreasing the risk of acute renal failure associated with HU [30]. When rasburicase was used at the initiation of chemotherapy, only 2% of paediatric patients with B-cell advanced-stage non-Hodgkin’s lymphoma (NHL) and L3 leukaemia required subsequent renal dialysis [31].

Our hypothesis was that this beneficial mechanism of rasburicase in the prevention of acute kidney injury (AKI) in TLS could be extrapolated to other aetiologies with HU, such as CN associated with acute CRS. To date, only one pilot study has been published on early treatment with rasburicase in patients undergoing cardiac surgery [5]. Twenty-six patients were randomized to receive rasburicase vs. placebo, and a single dose of rasburicase (7.5 mg) was administered intravenously 2 h before surgery. Rasburicase treatment did not improve renal function as measured by creatinine levels, but it improved urine neutrophil gelatinase-associated lipocalin (NGAL), an early marker of AKI. The authors found no differences in the need for vasopressor drugs between the two groups, and only one patient in the rasburicase group required RRT. Unlike our work, this study included patients with previous CKD (stage 3 according to the KDIGO classification) [42] and uric acid levels ≥6.5 mg/dL. Our study included a heterogeneous sample of patients with cardiac pathology and acute CRS with high diuretic requirements and significantly higher uric acid levels, which can probably explain why, haemodynamically, their response to rasburicase was more effective, achieving significant decreases in serum creatinine, eGFR and diuresis.

Although acute renal dysfunction in patients with decompensated heart failure is not well studied, it is known to be one of the main reasons for a prolonged hospital stay and worsening prognosis. Data from the ADHERE registry of more than 100,000 episodes of acute decompensated HF show that 45.5% of patients experience moderate renal dysfunction (stage G3), while 13.1% experience severe renal dysfunction (stage G4) and 7% experience AKI [35]. The presence of AKI increases mortality from 1.9% to 7%. Therefore, a therapeutic strategy that can help prevent renal damage and enhance diuretic response by decreasing congestion would be beneficial to improve the long-term prognosis in HF patients. In the present study, all patients treated with rasburicase experienced a significant diuretic response (2.800 cc per day on average), regardless of cardiac function and comorbidities. Consequently, congestion and NT-proBNP improved (from 24,293 to 6034 pg/mL), as did plasma UA (12.9 vs. 0.25 mg/dL) and creatinine (6 vs. 1.8 mg/dL). This increased diuretic response after a drastic decrease in UA levels may have been due to the fact that the transformation of UA into allantoin re-establishes urine flow, which may lead to post-obstructive polyuria. Segura et al. described the complete resolution of obstructive symptoms due to UA lithiasis after administration of two doses of rasburicase in two patients requiring haemodialysis [43].

At the time of the administration of rasburicase, patients were receiving high doses of diuretics and combinations of loop diuretics with thiazides and/or carbonic anhydrase inhibitors as standard treatment for CRS. These anionic diuretics reach the tubular lumen using URAT 1, NEFT 4 and OAT 1/3 [44,45]. UA uses these same transporters for its reabsorption and tubular secretion and thus competes with these diuretics, decreasing their response and favouring the intracellular and plasma accumulation of urate with its consequent toxicity.

In our study, all patients experienced increased diuresis after the administration of rasburicase and, consequently, the diuretic regimen was progressively reduced, with no patients requiring IV diuretics at the time of discharge. Our hypothesis is that the administration of rasburicase may help reverse diuretic resistance by eliminating UA, avoiding competition with diuretics upon their arrival at the tubular lumen. This recovers effective diuresis with a reduction in congestion, as shown by the reduction in Nt-proBNP at discharge in our patients.

The removal or uric acid with intermittent haemodialysis (IHD) proved to be effective in reversing hyperuricaemic acute renal failure in 11 out of 16 patients whose renal function rapidly returned to normal after dialysis in the study by Kjellstrand et al. [46].

Only five of our patients (14.2%) required RRT by CVVHF, with early withdrawal of RRT after 5, 10, 48, 72 and 98 h, respectively, due to increased diuresis. These patients were young (mean age: 56 vs. 70 years), and all were hospitalized at a CCU, suffering from haemodynamic instability, reflected by the need for inotropic drugs. This was the only aspect significantly associated with the need for RRT. It is difficult to extract conclusions from such a small sample, but this was the profile of those patients who would be candidates for heart transplantation should cardiac function not improve.

No association was found between RRT and the cardiac function parameters analyzed, such as NT-proBNP, LVEF, TAPSE and intravascular congestion measured by the absence of >50% collapse of the inferior vena cava. This suggests that early treatment with rasburicase improves the renal prognosis of these patients regardless of their cardiac function.

In the two biopsies performed, histology proved the absence of UA nephropathy, as well as changes related to uric acid haemodynamic effects, such as the presence of arteriolopathy of afferent arterioles and ischaemia phenomena. These were histological changes similar to those found by Roncal et al. in their cisplatin toxicity model with hyperuricaemia [47]. This would explain why UA reduction with rasburicase could improve renal function, reversing the acute haemodynamic effects and improving perfusion [48,49].

CRP is not only a marker of systemic inflammation but is also capable of decreasing endothelial nitric oxide (NO) synthesis [48]. Intracellular HU has been proved to stimulate CRP synthesis in both endothelial and vascular smooth muscle cells [50]. In our study, we obtained a significant reduction in CRP after administration of rasburicase, which could indicate that a negative uric acid balance can help reduce HU-associated inflammation.

This study has significant limitations. The number of patients was small, and we lacked a control group, so no definitive conclusions can be drawn. However, we believe that our results are relevant, as this is the first study to evaluate the beneficial effects of rasburicase in the treatment of acute CRS with HU. To date, there is no specific treatment for this syndrome, in which the deterioration in renal function negatively affects patients’ recovery and prognosis in terms of morbidity, duration of hospital stay and mortality.

All of our patients responded well to the treatment with rasburicase, with no clinically relevant adverse events.

However, these are preliminary results and further studies are needed to validate them, and a randomized controlled study is needed to confirm the potentially beneficial effect of rasburicase in the treatment of CRS.

## 4. Materials and Methods

### 4.1. Study Design

#### Single-Centre, Observational, Retrospective Study

Population: adult patients (age ≥ 18 years) treated with rasburicase for acute hyperuricaemia (>9 mg/dL) according to the rasburicase off-label use protocol approved by the Hospital’s Pharmacy and Therapeutics Committee from 2015 to 2023.

Patients were admitted to one of several cardiology units, including clinical cardiology and critical care [the coronary care unit (CCU) or the post-cardiac surgery intensive care unit (CICU)], and had a diagnosis of acute decompensated heart failure, according to the European Society of Cardiology (ESC) guidelines, and acute kidney injury stage II-II, according to KDIGO guidelines [29,42].

Patients were considered non-eligible for the administration of rasburicase in the following situations:History of allergy to rasburicase or any of its components;Confirmed or suspected glucose-6-phosphate dehydrogenase deficiency or other metabolic disorders causing haemolytic anaemia.

All hospitalized patients with acute CRS, decreased diuresis in spite of optimized diuretic treatment, worsening renal function and signs of systemic congestion were screened by the attending nephrologist. Those who fulfilled the criteria established in the protocol were administered a single dose of rasburicase intravenously.

The first 9 patients received a dose of 0.20 mg/kg, administered as a once daily 30 min intravenous infusion in 50 mL of a sodium chloride (9 mg/mL (0.9%)) solution, according to the manufacturer’s instructions. Afterwards, it was decided to use a standard dose of 6 mg for all patients, according to the recommendations of Mcdonnell et al., and to facilitate drug preparation [51]. All patients gave their informed consent.

### 4.2. Data Collection

The following variables were collected from the hospital records and the electronic prescription software:Demographics (age, sex, and weight);Underlying diseases: arterial hypertension (HTA), diabetes mellitus (DM), HF, rhabdomyolisis, kidney or heart transplant;Treatment: iodinated contrasts and/or vancomycin; anticalcineurin agents described as nephrotoxic drugs; use of inotropic drugs (dobutamine or levosimendan), vasopressors (noradrenaline, adrenaline or vasopressin) and diuretics (loop diuretics, thiazides, carbonic anhydrase inhibitors and aldosterone antagonists); and chronic treatment with xanthine oxidase inhibitors (XOis). Need for controlled ventilatory support (CVS) and/or ventricular assist devices (VADs) during hospital admission was also compiled;Renal function: serum creatinine and estimated glomerular filtration rate (eGFR) were calculated by the CKD-EPI equation at baseline, at 48–72 h after rasburicase administration, at hospital discharge and at least three months after hospital discharge. The need for RRT and its duration were recorded, as well as whether there was recovery of the diuretic rhythm, defined as a doubling of the diuretic rhythm with respect to baseline, or >100 cc/h;Cardiac function at the time of rasburicase: lactic acid, LVEF (preserved > 40%, reduced < 40%) and tricuspid annular systolic displacement (TAPSE) by echocardiography. Intravascular congestion was assessed by the presence or absence of >50% collapse of the inferior vena cava by ultrasound measurement and Nt-ProBNP levels at the time of rasburicase administration and at discharge;Inflammation and mortality: CRP levels were measured at the time of HU and maximum creatinine and at discharge. Intrahospital mortality rate and the causes of death were also assessed;Rasburicase adverse events (AEs), including allergy, headache, gastrointestinal symptoms, fever, skin rash, haemolysis, haemolytic anaemia, metahaemoglobinaemia and seizures.

Renal biopsies: Although histological studies were not considered as part of the study, kidney biopsies were performed in two patients for clinical reasons not related to CRS and the decision to use rasburicase. Two cylinders were extracted for study in light microscopy (LM) and immunofluorescence (IF). For LM, the tissue was fixed in formalin with standard histochemical stains: haematoxylin–eosin stain (HE), periodic acid–Schiff (PAS), Jones methenamine silver and trichrome. For IF, standard immunohistochemical studies with IgA, IgG, IgM, k, ʎ, C3, C1q, albumin and fibrin were used.

### 4.3. Statistical Analysis

Statistical analysis was performed using SPSS21.0 software (SPSS, Chicago, IL, USA). Qualitative variables are presented with their frequency distributions. Quantitative variables are presented with their means and standard deviations (SDs) or medians and interquartile ranges (IQRs) according to their distributions. The association between qualitative variables was evaluated with the χ^2^ (chi square) test or Fisher’s exact test. Quantitative variables were analyzed using the Student’s *t*-test (in comparisons of one variable with two categories) if the variables followed a normal distribution and the Wilcoxon or Mann–Whitney test in the case of variables with a non-parametric distribution. The distribution of the variables was verified with the theoretical models, and the hypothesis of homogeneity of variance was tested. Statistical significance was considered as a *p* value < 0.05.

## 5. Conclusions

Our study shows that a single dose of rasburicase could re-establish the diuretic response and improve congestive nephropathy associated with CRS, reducing the need for RRT and its duration and improving the prognosis of patients. Blinded randomized controlled trials with larger sample sizes will be necessary to validate these promising results.

## Figures and Tables

**Figure 1 ijms-25-03329-f001:**
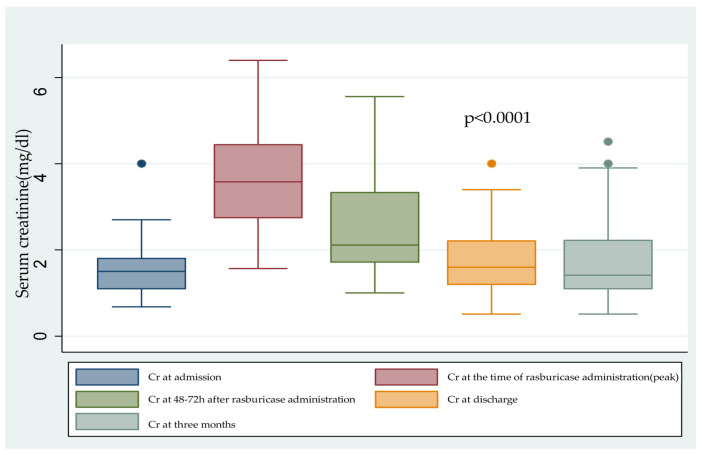
Variability in creatinine progression before and after rasburicase treatment.

**Figure 2 ijms-25-03329-f002:**
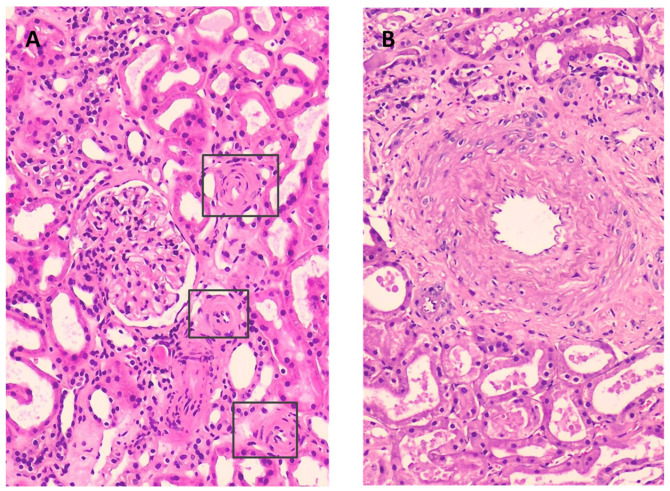
Photomicrograph of renal biopsies in two of the patients. (**A**) Renovascular hypertension. Small-calibre arteries with wall thickening at the expense of concentric myointimal hyperplasia with occlusion of 70% of the lumen are marked. Interstitium: tubules with preserved epithelial lining, without crystalloid elements in the lumen, and ischaemic-looking glomerulus (HE, 10×). (**B**) Nephroangiosclerosis. Small-calibre vessel with intimal fibroelastosis occluding 50% of the lumen (HE, 20×).

**Table 1 ijms-25-03329-t001:** Patient characteristics.

**Demographics**	
Age (years), mean (SD)	67.5 (15.4)
Sex, male, n (%)	23 (65.7)
**Underlying diseases**	
Arterial hypertension, n (%)	30 (85.7)
Diabetes mellitus, n (%)	13 (37.1)
Heart failure, n (%)	34 (97.1)
Rhabdomyolysis, n (%)	4 (11.4)
Kidney or heart transplant, n (%)	8 (22.8)
Cr at admission (mg/dL), mean (SD)	1.58 (0.68)
eGFR CKD-EPI at admission (mL/min/1.73 m^2^), mean (SD)	43.41 (19)
**Treatment**	
Contrast, n (%)	11 (31.4)
Nephrotoxic drugs, n (%)	21 (60)
Inotropic drugs, n (%)	15 (42.8)
Vasopressor drugs, n (%)	16 (45.7)
Loop diuretics, n (%)	35 (100)
Two or more diuretics, n (%)	16 (45.7)
XOis, n (%)	13 (37.14)
CVS, n (%)	6 (16.7)
VAD, n (%)	6 (16.7)
Renal function	
AKI II, n (%)	18 (51.4)
AKI III, n (%)	17 (48.6)
RRT, n (%)	5 (14.2)
Cardiac function	
Lactic acid (mg/dL), mean (SD)	1.55 (0.4)
TAPSE (cm) (SD)	17.3 (5.1)
LVEF < 40%, n (%)	15 (42.8)
Absence of >50% collapse of the inferior vena cava, n (%)	27 (77.1)

eGFR: Estimated Glomerular Filtration Rate; SD: standar deviation, XOis: xanthine oxidase inhibitors; CVS: controlled ventilatory support; VAD: ventricular assist device; Cr: creatinine; CKD-EPI: Chronic Kidney Disease Epidemiology Collaboration; AKI: acute kidney injury; RRT: renal replacement therapy; TAPSE: tricuspid annular systolic displacement; LVEF: Left Ventricular Ejection Fraction.

**Table 2 ijms-25-03329-t002:** Evolution of renal and cardiac function parameters after rasburicase administration.

	Value at the Time of Rasburicase Administration (Peak)	Value at Discharge	*p*
Cr (mg/dL), mean (SD)	3.65 (1.27)	1.8 (0.8)	<0.0001
eFGR CKD-EPI (mL/min/1.73 m^2^), mean (SD)	17 (8)	41 (20)	<0.0001
Nt-proBNP (pg/mL), median (IQR)	24,298 (7122–35,000)	6034 (1973–11,191)	<0.0001
CRP (mg/L), median (IQR)	10 (3.4–39.27)	4.3 (1.0–10.2)	<0.0001

Cr: creatinine; eGFR CKD-EPI: estimated Glomerular Filtration Rate Chronic Kidney Disease Epidemiology Collaboration; Nt-proBNP: N-terminal pro b-type natriuretic peptide; CRP: C-Reactive protein.

**Table 3 ijms-25-03329-t003:** Comparison of patients who needed RRT with those who did not.

	RRT (n = 5)	No RRT (n = 30)	*p*
**Age (years), mean (SD)**	56 (16)	70 (14)	0.047
**CRP (mg/L), median (IQR)**	80.5 (1.6–186)	10.9 (3.5–29.4)	ns
**eGFR CKD-EPI (mL/min/1.73 m^2^), mean (SD)**	49.6 (14)	43 (20)	ns
**Arterial hypertension, n (%)**	4 (80)	26 (86.6)	ns
**Diabetes mellitus, n (%)**	2 (40)	11 (36)	ns
**LVEF (%), mean (SD)**	42 (13)	39 (14)	ns
**TAPSE (cm), mean (SD)**	16.2 (4.34)	17.5 (5.4)	ns
**Absence of >50% collapse of the inferior vena cava, n (%)**	5 (18.5)	22 (81.5)	ns
**Lactic acid (mg/dL), mean (SD)**	1.7 (0.5)	1.5 (0.4)	ns
**Nt-proBNP (pg/mL), median (IQR)**	24,429 (9201–35,000)	24,158 (6806–35,000)	ns
**VAD, n (%)**	2 (40)	4 (13.3)	ns
**CVS, n (%)**	2 (40)	4 (13.3)	ns
**Inotropic drugs, n (%)**	5 (100)	10 (33.3)	0.009

VAD: ventricular assist device; CVS: controlled ventilatory support; CKD-EPI: Chronic Kidney Disease Epidemiology Collaboration; RRT: renal replacement therapy; TAPSE: tricuspid annular systolic displacement.

**Table 4 ijms-25-03329-t004:** Conditions that favour severe HU in CRS.

Conditions That Favour Severe HU in CRS
Hypoxemia
ATP degradation
Decrease in glomerular filtration rate
Activation of the renin–angiotensin–aldosterone system
Decrease in effective circulating volume
Diuretics
Acetylsalicylic acid
Cyclosporine and tacrolimus
Lactic acid production and ketoacidosis
Surgery with general anaesthesia
Post-surgery rewarming
Ischaemia–reperfusion
Haemolysis (VAD, BiAo, and Impella)
Rhabdomyolysis

BiAo: intra-aortic balloon.

## Data Availability

The data presented in this study are available on request from the corresponding author. The data are not publicly available due to privacy or ethical restrictions.

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
