# Peer review of "Treatment with Rasburicase in Hospitalized Patients with Cardiorenal Syndrome: Old Treatment, New Scenario"

_ijms, 2024, doi:10.3390/ijms25063329_

Round 1

Reviewer 1 Report

Comments and Suggestions for Authors

See in attachment

Author Response

We thank reviewer for the comments.  We agree that a controlled study is needed to confirm our preliminary results and will try to address this in our future research. This limitation has been added to the discussion:  

A randomised controlled study is needed to confirm the potentially beneficial effect of rasburicase in the treatment of CRS. 

As mentioned, in our study 13 patients (37%) were diabetic. We totally agree with the reviewer that SGLT2 inhibitors by increasing glucosuria could also play a role decreasing plasma urate.  

The following paragraph and the suggested reference have been added to the discussion: 

In regards to hyperuricemia, SGLT2 inhibitors can also play a relevant role. In patients with type 1 diabetes, Lytvyn et al found that by increasing glucosuria, SGLT2i can provide a 10 to 15% reduction in plasma urate because of uric acid secretion in exchange for glucose reabsorption via the GLUT9 transporter (41). This possible relationship between plasma uric acid levels and SLGT2-I associated uricosuria may be beneficial to prevent CRS although clinical relevance is not yet established. However, the initiation of SGLT2i is not currently recommended in situations of acute renal dysfunction. Manufacturer´s advise that dapagliflozin treatment should not be initiated in patients with GFR < 25 ml/min and empagliflozin in patients with GFR <20 ml/min due to limited experience. Their relevance in CRS should be addressed in future research. On the other hand, rasburicase can be used safely in acute renal failure to prevent further deterioration.  

Reviewer 2 Report

Comments and Suggestions for Authors

Comments to authors: 

There are two forms of urate nephropathy:  one in which there is over- production of urate such as in tumor lysis syndrome (TLS) wherein uric acid excretory burden is markedly increased resulting in precipitation of urate crystals in the distal nephron, a form of intrarenal obstructive nephropathy; the second form of urate nephropathy is one where the disorder is mainly one of reduced urate excretion, sometimes with some increased production, wherein  renal tubular obstruction is not the mechanism of eventual renal disease.  Cardiorenal syndrome is in the second category wherein the hyperuricemia is caused by decreased GFR, caused by heart failure or renal vascular disease,  decreased secretion of uric acid due to competition by diuretics for the organic acid transport system, and to enhanced absorption of filtered urate due to markedly increased proximal tubular fluid absorption caused by heart failure.   The patients studied in the present manuscript belong in the second group.   On the other hand, the drug used, rasburicase, was intended for managing conditions with markedly increased uric acid production.  The two biopsies reported in the present manuscript attest to the fact that the patients in this study belong in the second group as no evidence of tubular urate precipitates was observed and instead severe blood vessel disease was noted.   Thus it would have been cogent to have demonstrated that conventional uric acid lowering therapy by xanthine oxidase (XO)  inhibitors or uricosuric drugs was not sufficient to manage these patients with cardiorenal syndrome.  The data presented by the authors is very interesting in that it purports to shows prompt improvement in urine output, renal function, a decrease in BNP and CRP.  However it would have been well to have shown that these patients had failed therapy with XOs or uricosurics before embarking on this novel therapeutic approach.

Specific comments/criticisms:

1)      The authors need to show data of urine output before and after initiation of rasburicase.

2)      Authors need to indicate whether all or some of these patients were ever tried on XOs or uricosuric drugs

3)      What was the indication  for the two renal biopsies ? and how did the information derived from them enlighten the therapeutic approach to the patients?

MINOR POINT/S

LINE  260: refers to the lack of association with intravascular congestion.   How was “intravascular congestion” defined

LINE 45  :  is this a typo :HA? 

Line 162:  what is “CMV” 

LINE 247   do you mean tubular lumen? “in their arrival at the luminal lumen” 

LINE 243:   “In our study, all patients increased their diuretic rhythm”  what is diuretic rhythm?

Line 230: correct highlighted data re plasma uric acid.   Consequently, congestion and NT-proBNP improved (from 24,293 to 6,034 pg/ml), as did plasma UA (0.25 vs 12.9 mg/dl

Comments on the Quality of English Language

see above 

Author Response

Point-by-point response to Reviewer´s Comments and Suggestions 

  1. The authors need to show data of urine output before and after initiation of rasburicase. 

Second paragraph in Results section 2.1 has been modified as follows: 

All patients (100%) achieved diuretic response between 12 and 98h after rasbu-ricase administration with a mean diuresis of 2800±1200 cc /day compared to a mean diuresis of 1540± 620cc/day (p<0.0001) before rasburicase administration. In those patients receiving IV furosemide combined with other diuretics (34.3%) the mean diuretic response after rasburicase administration was 4.200± 1051cc/day. 

2. Authors need to indicate whether all or some of these patients were ever tried on XOs or uricosuric drugs.  

We thank the reviewer for his/her comments regarding xanthine oxidase inhibitors. We have now included the number of patients that were receiving xanthine oxidase inhibitors as part of their chronic treatment. Methods section and table 1 have been modified accordingly. The following paragraph has been added to the discussion: 

In our cohort, 37.4% of the patients were receiving xanthine oxidase inhibitors (XOi) as part of their chronic treatment. Although they could potentially help prevent severe hyperuricemia, XOi exert their action by inhibiting the de novo formation of uric acid thus maintaining the renal toxicity of the xanthine crystals that are formed by inhibiting xan-thine oxidase, whereas rasburicase is capable of eliminating the preformed uric acid, avoiding its deleterious effects and preventing the formation of xanthine crystals. On the other hand uricosuric drugs (e.g. lesinurad, benzbromanona, and probenecid) are contra-indicated in patients with severe renal failure. For this reason, we decided to treat our patients with rasburicase and not use xanthine oxidase inhibitors or uricosurics in the acute phase of CRS. 

3. What was the indication for the two renal biopsies ? and how did the information derived from them enlighten the therapeutic approach to the patients? 

The two renal biopsies were considered appropriate for clinical reasons non-related to CRS and the decision to use rasburicase.  Methods section and results have being modified to include this statement. 

The first patient was in the in the second month after receiving a heart transplant, and was being treated with tacrolimus, ganciclovir and trimethoprim- sulfamethoxazole. The patient had suffered primary graft failure, so it was not appropriate to reduce immunosuppressant dose, with tacrolimus levels being at that time around at 9ng/ml. Patient developed hypertension that proved difficult to control, with rapidly progressive deterioration of renal function. Hence, the biopsy was performed to rule out renal thrombotic microangiopathy associated with tacrolimus, or any type of recent glomerular involvement. The result of the biopsy ruled out renal thrombotic microangiopathy and helped us maintain the heart transplant treatment.  

The second biopsy corresponds to a patient with a ventricular assist device who died after its removal. Therefore, the biopsy results were merely educational for our daily clinical practice. 

MINOR POINT/S 

LINE  260: refers to the lack of association with intravascular congestion.   How was “intravascular congestion” defined 

As already established in the methods section, intravascular congestion was assessed by the presence or absence of >50% collapse of the inferior vena cava. This was measured by ultrasound techniques. The technique was previously referred to as echography, and we have realised this may not be the right English translation, so it has been modified. 

We have now included this parameter in table 3, to show that no differences were found between patients with and without RRT and  the information that was in line 260 has been modified. For better clarification, now the paragraph states:  

No association was found between RRT and the cardiac function parameters ana-lysed, such as NT-proBNP, LVEF, TAPSE and intravascular congestion measured by absence of >50% collapse of the inferior vena cava.  

LINE 45  :  is this a typo :HA? Yes, it meant to say HU as it stands for hyperuricemia. It has been corrected. We thank the reviewer for noticing.  

Line 162:  what is “CMV” . This is a typo, we apologize. It meant to say CVS, standing for controlled ventilatory support. We have corrected it in the article.  

LINE 247   do you mean tubular lumen? “in their arrival at the luminal lumen” . We did, again we apologize for the mistake and have corrected it.  

LINE 243:   “In our study, all patients increased their diuretic rhythm”  what is diuretic rhythm? 

We meant to say that all patients significantly increased diuresis after the administration of rasburicase. We have replaced diuretic rhythm with increased diuresis. 

Line 230: correct highlighted data re plasma uric acid.   Consequently, congestion and NT-proBNP improved (from 24,293 to 6,034 pg/ml), as did plasma UA (0.25 vs 12.9 mg/dl 

It has been corrected, thank you. 

We thank reviewer 2 for his/her suggestions and for detecting our errors. English language has also been revised as per reviewer´s suggestion. 
